# Validation of a Deep Learning Chest X-ray Interpretation Model: Integrating Large-Scale AI and Large Language Models for Comparative Analysis with ChatGPT

**DOI:** 10.3390/diagnostics14010090

**Published:** 2023-12-30

**Authors:** Kyu Hong Lee, Ro Woon Lee, Ye Eun Kwon

**Affiliations:** Department of Radiology, College of Medicine, Inha University, Incheon 22212, Republic of Korea

**Keywords:** ChatGPT, KARA-CXR, chest X-ray, LLM

## Abstract

This study evaluates the diagnostic accuracy and clinical utility of two artificial intelligence (AI) techniques: Kakao Brain Artificial Neural Network for Chest X-ray Reading (KARA-CXR), an assistive technology developed using large-scale AI and large language models (LLMs), and ChatGPT, a well-known LLM. The study was conducted to validate the performance of the two technologies in chest X-ray reading and explore their potential applications in the medical imaging diagnosis domain. The study methodology consisted of randomly selecting 2000 chest X-ray images from a single institution’s patient database, and two radiologists evaluated the readings provided by KARA-CXR and ChatGPT. The study used five qualitative factors to evaluate the readings generated by each model: accuracy, false findings, location inaccuracies, count inaccuracies, and hallucinations. Statistical analysis showed that KARA-CXR achieved significantly higher diagnostic accuracy compared to ChatGPT. In the ‘Acceptable’ accuracy category, KARA-CXR was rated at 70.50% and 68.00% by two observers, while ChatGPT achieved 40.50% and 47.00%. Interobserver agreement was moderate for both systems, with KARA at 0.74 and GPT4 at 0.73. For ‘False Findings’, KARA-CXR scored 68.00% and 68.50%, while ChatGPT scored 37.00% for both observers, with high interobserver agreements of 0.96 for KARA and 0.97 for GPT4. In ‘Location Inaccuracy’ and ‘Hallucinations’, KARA-CXR outperformed ChatGPT with significant margins. KARA-CXR demonstrated a non-hallucination rate of 75%, which is significantly higher than ChatGPT’s 38%. The interobserver agreement was high for KARA (0.91) and moderate to high for GPT4 (0.85) in the hallucination category. In conclusion, this study demonstrates the potential of AI and large-scale language models in medical imaging and diagnostics. It also shows that in the chest X-ray domain, KARA-CXR has relatively higher accuracy than ChatGPT.

## 1. Introduction

Artificial intelligence (AI) revolutionizes healthcare by improving clinical diagnosis, administration, and public health infrastructures. AI applications in healthcare include disease diagnosis, drug discovery, assisted surgeries, and patient care. AI can enhance healthcare outcomes, reduce costs, and optimize treatment planning [1]. However, challenges to be overcome include ensuring ethical boundaries, addressing bias in AI algorithms, and maintaining diversity, transparency, and accountability in algorithm development. AI is not meant to replace doctors and healthcare providers but to complement their skills through human—AI collaboration. The human-in-the-loop approach ensures safety and quality in healthcare services, where AI systems are guided and supervised by human expertise. 

The rise in large language models (LLMs) in AI has garnered significant attention and investment from companies like Google, Amazon, Facebook, Tesla, and Apple. LLMs, such as OpenAI’s GPT series and ChatGPT, have shown remarkable progress in tasks like text generation, language translation, and question answering. These models are trained on massive amounts of data and have the potential to display intelligence beyond their primary task of predicting the next word in a text.

LLMs have the potential to revolutionize healthcare by assisting medical professionals with administrative tasks, improving diagnostic accuracy, and engaging patients [2]. LLMs, such as GPT-4 and Bard, can be implemented in healthcare settings to facilitate clinical documentation, obtain insurance pre-authorization, summarize research papers, and answer patient questions [3]. They can generate personalized treatment recommendations, laboratory test suggestions, and medication prompts based on patient information [4]. It is essential to ensure LLMs’ responsible and ethical use in medicine and healthcare, considering privacy, security, and the potential for perpetuating harmful, inaccurate, race-based content [5]. LLMs, like ChatGPT, can accelerate the creation of clinical practice guidelines by quickly searching and selecting evidence from numerous databases [6]. 

KakaoBrain AI for Radiology Assistant Chest X-ray (KARA-CXR) is a new medical technology that helps in radiological diagnosis. Developed by leveraging the cutting-edge capabilities of large-scale artificial intelligence and advanced language models, this cloud-based tool represents a significant leap in medical imaging analysis. The core functionality of KARA-CXR lies in its ability to generate detailed radiological reports that include findings and conclusions. This process is facilitated by its sophisticated AI, which has been trained on vast datasets of chest X-ray images. By interpreting these images, KARA-CXR can provide accurate and swift diagnostic insights essential in clinical decision-making. 

Based on the GPT-4V architecture, ChatGPT has potential in the medical field, especially for interpreting chest X-ray images. This language model can analyze medical images, including chest X-ray data, to generate human-like reading reports. Although not yet available for clinical use, by providing a general interpretation of chest X-rays, ChatGPT has the potential to improve the diagnostic process, especially in settings with limited access to radiology expertise [7]. In this study, we analyze the diagnostic accuracy and utility of KARA-CXR and ChatGPT and discuss their potential for use in clinical settings.

## 2. Materials and Methods

### 2.1. Dataset

We randomly selected 2000 chest X-ray images (PA projection) from a single institution (Inha University Hospital, Incheon, Republic of Korea) from 2010 to 2022. The selected images were all of Asian individuals, with a male and female ratio of 46% and 54%, respectively. To ensure ease of reading, we excluded pediatric patients, poor-quality images (images not taken with digital equipment or taken with portable equipment) and selected only images of adult patients (aged 19 to 99 years, median age: 46.8 ± 2.5 SD). Furthermore, to ensure fairness in the assessment of reading difficulty, percentages were not separately established for each image’ disease. Finally, the examination of the selected images and the decision to include them in the dataset was performed by a radiologist with 10 years of experience (Ro Woon, Lee). Furthermore, all included images were fully anonymized using Python (version 3.12) and then serially numbered for analysis. The files were exported as DICOM files.

### 2.2. Input Data

Anonymized DICOM files were uploaded to both KARA-CXR (Kakaobrain, Seoul, Republic of Korea) and ChatGPT (OpenAI, San Francisco, CA, USA). To anonymize medical images for analysis in ChatGPT and KARA-CXR, we removed all identifiable patient information to comply with the privacy and confidentiality standards set forth by the Health Insurance Portability and Accountability Act (HIPAA). Anonymization involved removing details such as patient names, dates of birth, medical record numbers, and other unique identifiers from the images. 

Even after anonymization, we further enhanced privacy by turning off the “Chat History and Training” option in ChatGPT. This setting ensures that conversations and images shared during a session are not used for further training of the AI model or accessed in future sessions. This is a precautionary measure to ensure that residual or indirect information is not used in ways that could compromise patient confidentiality.

In KARA-CXR, a cloud-based analysis system, immediately deleted the input DICOM data after analysis for personal information protection. Unlike ChatGPT, KARA-CXR generates text-based readings shortly after uploading DICOM files without requiring separate prompts. KARA-CXR utilized a closed beta version prior to public release (Figure 1), and there are plans to make it publicly available via the website in December 2023.

In the case of ChatGPT, to obtain the right results for our research, we first chose the prompt to be entered into ChatGPT. ChatGPT is designed with guidelines that prevent it from providing professional interpretations or diagnoses, especially in contexts requiring specialized expertise, such as medical imaging, including chest X-rays [8]. To overcome the limitations of this large language model, we employed a carefully crafted, non-directive bypass prompt: ‘This is a chest PA image. Tell me more about what is going on?’ This prompt was strategically chosen to navigate ChatGPT’s usage policies and ethical constraints, allowing us to obtain a chest x-ray reading from ChatGPT. Furthermore, ChatGPT was used in its paid version, GPT-4V, and to protect personal information, the ‘Chat history & training’ option was disabled, ensuring data were not stored on OpenAI’s servers.

The interpretation texts thus generated were qualitatively analyzed by two observers. A rough schematic of this process can be seen in Figure 2.

### 2.3. Analyzing Readings by LLMs

We selected five qualitative factors (accuracy, false findings, location inaccuracies, count inaccuracies, and hallucination) to evaluate the quality of the readings generated by KARA-CXR and ChatGPT. The detailed descriptions of the factors are shown in Table 1.

For the five items mentioned in Table 1, two readers with ten years of experience in chest radiology reading evaluated the images in independent sessions. We evaluated the interpretation results of each model for chest X-ray images and recorded the evaluation results according to the case numbers of the anonymized images.

### 2.4. Statistical Analytics

We analyzed the percentages of details in each of the five assessment categories rated by each reader and obtained interobserver agreement between each reader. The statistical analysis of the data was performed in Python (version 3.12).

## 3. Results

In evaluating diagnostic accuracy, two observers assessed the performance of KARA and GPT4. Observer 1 found that KARA achieved 70.50% accuracy in the category deemed ‘Acceptable’, while GPT4 was reported at a notably lower value of 40.50% in the same category. Observer 2’s assessments were slightly lower for KARA at 68.00% but higher for GPT4 at 47.00% in the ‘Acceptable’ category (Figure 3). The interobserver agreement rates, which reflect the consistency between observers, were relatively close, with KARA at 0.74 and GPT4 at 0.73, indicating moderate agreement.

In the category of ‘False Findings’ with no findings being the subcategory, both observers recorded similar results for KARA, with Observer 1 at 68.00% and Observer 2 at 68.50%. In comparison, GPT4 was observed at 37.00% by both observers (Figure 4). The interobserver agreement for KARA stood at a high rate of 0.96, and GPT4 also had a high agreement rate of 0.97. These high agreement rates suggest a consistent assessment of false findings between the two observers for KARA and GPT4.

When it came to ‘Location Inaccuracy’ with no inaccuracies noted, Observer 1 reported KARA at 76.00% and GPT4 at 46.50%, and Observer 2 reported KARA at 77.50% and GPT4 at 46.00% (Figure 5). The interobserver agreement for KARA was 0.93, indicating high consistency, whereas GPT4’s agreement was lower at 0.83, signifying a moderate-to-high consistency between observers.

‘Count Inaccuracy’ with no inaccuracies was observed at a high rate by both observers for KARA (94.00%) and GPT4 (90.00%), reflecting a very high level of performance in this category (Figure 6). The interobserver agreement for KARA and GPT4 was at 0.99, indicating almost perfect consistency between the two observers.

Lastly, the ‘Hallucination’ category with no instances reported by Observer 1 showed KARA at 75.88% and GPT4 at 38.69%, while Observer 2 reported the same percentage for KARA and a slightly lower 38.19% for GPT4 (Figure 7). The interobserver agreement was 0.91 for KARA and 0.85 for GPT4, demonstrating high consistency for KARA assessments and moderate-to-high consistency for GPT4.

## 4. Discussion

AI is being increasingly used in the field of chest X-ray reading. It has various applications, including lung cancer risk estimation, detection, and diagnosis, reducing reading time, and serving as a second ‘reader’ during screening interpretation [9]. Doctors in a single hospital reported positive experiences and perceptions of using AI-based software for chest radiographs, finding it useful in the emergency room and for detecting pneumothorax [10]. A model for automatic diagnosis of different diseases based on chest radiographs using machine learning algorithms has been proposed [11]. In a multicenter study, AI was used as a chest X-ray screening tool and achieved good performance in detecting normal and abnormal chest X-rays, reducing turnaround time, and assisting radiologists in assessing pathology [12]. AI solutions for chest X-ray evaluation have been demonstrated to be practical, perform well, and provide benefits in clinical settings [13].

However, conventional labeling-based chest X-ray reading AI has limitations in terms of accuracy and efficiency. The manual labeling of large datasets is expensive and time-consuming. Automatic label extraction from radiology reports is challenging due to semantically similar words and missing annotated data [14]. In a multicenter evaluation, the AI algorithm for chest X-ray analysis showed lower sensitivity and specificity values during prospective validation compared to retrospective evaluation [15]. However, the AI model performed at the same level as or slightly worse than human radiologists in most regions of the ROC curve [15]. A method for standardized automated labeling based on similarity to a previously validated, explainable AI model-derived atlas has been proposed to overcome these limitations. Fine-tuning the original model using automatically labeled exams can preserve or improve performance, resulting in a highly accurate and more generalized model.

The effectiveness of deep-learning based computer-aided diagnosis has been demonstrated in disease detection [16]. However, one of the major challenges in training deep learning models for medical purposes is the need for extensive, high-quality clinical annotation, which is time-consuming and costly. Recently, CLIP [17] and ALIGN [18] have shown the ability to perform vision tasks without any supervision. However, vision-language pre-training (VLP) in the CXR domain still lacks sufficient image-text datasets because many public datasets consist of image-label pairs with different class compositions.

The rise in medical image reading with large language models has gained significant attention in recent research [19]. Language models have been explored to improve various tasks in medical imaging, such as image captioning, report generation, report classification, finding extraction, visual question answering, and interpretable diagnosis. Researchers have highlighted the potential benefits of accurate and efficient language models in medical imaging analysis, including improving clinical workflow efficiency, reducing diagnostic errors, and assisting healthcare professionals in providing timely and accurate diagnoses [20]. 

KARA-CXR is an innovative cloud-based medical technology that utilizes artificial intelligence and advanced language models to revolutionize radiological diagnostics. It operates over the web and offers a user-friendly interface for healthcare professionals. KARA-CXR generates detailed radiological reports with findings and conclusions by analyzing chest X-ray images uploaded in DICOM format. This is made possible by its sophisticated AI, which has been trained on vast datasets of chest X-ray images. The technology provides accurate and swift diagnostic insights, aiding radiologists in ensuring precise diagnoses and reducing report generation time. KARA-CXR is particularly valuable in high-volume or resource-limited settings where radiologist expertise may be scarce or overburdened.

In this study, ChatGPT based on GPT-4V architecture showed some potential in interpreting chest X-ray images but also revealed some limitations. ChatGPT can generate human-like diagnostic reports based on chest X-ray data through extensive reinforcement learning on the medical text and imaging data included during development. However, due to the limitations of reinforcement learning based on information openly available on the internet, we must recognize that the data generated by ChatGPT do not guarantee medical expertise. In conclusion, it is essential to note that ChatGPT is not a substitute for professional medical advice, diagnosis, or treatment [21]. 

In our study, detailed observations of reports indicate that KARA generally outperforms GPT4 across various categories of diagnostic accuracy, with consistently higher percentages and interobserver agreement rates. The data suggest a significant discrepancy between the two systems, with KARA displaying more reliable and accurate performance as per the observers’ evaluations. Particularly in terms of hallucination, KARA-CXR demonstrated superior performance compared to ChatGPT. ChatGPT sometimes produced incorrect interpretation results, including hallucinations, even in cases with clinically significant and obvious abnormalities such as pneumothorax (Figure 8).

In our comparative analysis between KARA-CXR and ChatGPT, a striking advantage of KARA-CXR was observed in the hallucination. Notably, KARA-CXR demonstrated a significantly higher percentage in non-hallucinations with a non-hallucination rate of 75% as compared to that of only 38% for ChatGPT, as agreed upon by both observers. This substantial difference underscores the superior capability of KARA-CXR in providing reliable and accurate interpretations in chest X-ray diagnostics, a crucial aspect in the field of medical imaging where the precision of diagnosis can significantly impact patient outcomes. The propensity of ChatGPT to generate more hallucinations in medical contexts can be attributed to its foundational design and training methodology. As a large language model, ChatGPT is trained on a vast corpus of text from diverse sources, not specifically tailored for medical diagnostics. This generalist approach, while versatile, can lead to inaccuracies and hallucinations, especially in highly specialized fields like medical imaging [22]. Despite its potential, the accuracy and reliability of ChatGPT responses should be carefully assessed, and its limitations in understanding medical terminology and context should be addressed [23]. In contrast, KARA-CXR, designed explicitly for medical image analysis, benefits from a more focused training regime, enabling it to discern nuanced details in medical images more effectively and reducing the likelihood of generating erroneous interpretations.

In our exploration of ChatGPT’s application to medical imaging, particularly in chest X-ray interpretation, a notable limitation emerged, meriting explicit mention. ChatGPT, in its current design, is programmed to refuse direct requests for the professional interpretation of medical images, such as X-rays [8]. This usage policy and ethical boundary, built into ChatGPT to avoid the non-professional practice of medicine, significantly impacts its clinical application in this context. In the initial process of our study, we observed that direct prompts requesting chest X-ray interpretation were consistently declined by ChatGPT, aligning with its programming to avoid assuming the role of a radiologist or other medical professional. This limitation is critical to understand for any future research utilizing ChatGPT or similar language models in medical image interpretation. Despite the impressive capabilities of AI in healthcare, such as KARA-CXR and ChatGPT, hallucinations can cause serious problems in real-world clinical applications of AI. Such hallucinations may be of minimal consequence in casual conversation or other contexts but can pose significant risks when applied to the healthcare sector, where accuracy and reliability are of paramount importance. Misinformation in the medical domain can lead to severe health consequences on patient care and outcomes. The accuracy and reliability of information provided by language models can be a matter of life or death. They pose real-life risks, as they could potentially affect healthcare decisions, diagnosis, and treatment plans. Hence, the development of methods to evaluate and mitigate such hallucinations is not just of academic interest but of practical importance.

While promising, the integration of SaMD (software as medical device), including KARA-CXR and ChatGPT, into medical diagnostics faces several challenges that must be addressed in future research. A primary concern is the diversity of data used to train these AI models. Often, AI systems are trained on datasets that may only adequately represent some population groups, leading to potential biases and inaccuracies in diagnostics, particularly for underrepresented demographics. Moreover, these AI systems’ “black box” nature poses a significant challenge. The internal mechanisms of how they analyze and interpret chest X-ray images are only partially transparent, making it difficult for healthcare professionals to understand and thus trust the conclusions drawn by these technologies.

Another notable limitation is the integration of these AI tools into clinical practice. Healthcare professionals may be hesitant to depend on AI for critical diagnostic tasks due to concerns about the accuracy and reliability of these systems, as well as potential legal and ethical implications. Building trust in AI technologies is essential for their successful adoption in medical settings [24]. In addition to these concerns, it is essential to keep in mind that even if an AI-powered diagnostic solution is highly accurate, the final judgment should still be made by a medical professional—a doctor.

To overcome these challenges, future research should focus on enhancing the diversity of training datasets, including a broader range of demographic data, to ensure that AI models can deliver accurate diagnostics across different populations [25]. It is also crucial to improve the transparency and explainability of AI algorithms, developing methods to demystify the decision making process and increase acceptability and trustworthiness among medical practitioners [26]. Although this paper evaluates the diagnostic accuracy of two potential SaMDs, ChatGPT and KARA-CXR, one of the limitations is that there needs to be a clear rationale or recommendation for evaluating or approving such software, legally or within the academic community. The limitation in approving software for medical use (SaMD) stems from the need for a clear definition of SaMD, which makes it difficult to create standards and regulations for its development and implementation [27]. Without clear boundaries, there are risks to patient safety because not all components potentially impacting SaMD are covered by regulations [27]. This lack of clarity also affects innovation and design in the field of SaMD, as new technology applications that support healthcare monitoring and service delivery may need to be more effectively regulated [28]. We believe that gradually, along with software development, we will need to establish factors and regulations that will define the clinical accuracy and safety of these SaMDs. Extensive clinical validation studies are necessary to establish the reliability and accuracy of AI-based diagnostic tools, adhering to high ethical standards and regulatory compliance [29]. These studies should also address patient privacy, data security, and the potential ramifications of misdiagnoses [29]. By focusing on these areas, the potential of AI in medical diagnostics can be more fully realized, leading to enhanced patient care and more efficient healthcare delivery.

The limitations of this study include that this research was conducted as a single-institution study, which presents certain limitations. One of the primary constraints was the limited number of images that could be analyzed due to the restricted number of researchers involved in the study. This limitation could potentially impact the generalizability of our findings to broader image populations and diverse clinical settings. Another significant limitation was the lack of a reference standard for the chest X-ray interpretations. Although we analyzed the interobserver agreement between the readers, the absence of a definitive standard means that even interpretations by experienced readers cannot be considered definitive answers. This aspect could affect the reliability and validity of the diagnostic conclusions drawn in our study. Additionally, ethical considerations programmed into ChatGPT led to the refusal of direct requests for chest image interpretation, necessitating the use of indirect prompts to obtain diagnostic interpretations. This workaround might have influenced the quality and accuracy of the results derived from ChatGPT. We acknowledge that the possibility of obtaining more accurate results from ChatGPT cannot be entirely ruled out if direct requests for chest X-ray interpretation were permissible.

## 5. Conclusions

This study underscores the potential of AI in improving medical diagnostic processes, with specific emphasis on chest X-ray interpretation. While KARA demonstrates superior precision in image analysis, ChatGPT excels in contextual data interpretation. The key takeaway is the complementary nature of these technologies. A hybrid approach, integrating KARA’s imaging expertise with ChatGPT’s comprehensive analysis, could lead to more accurate and efficient diagnostic processes, ultimately improving patient care.

The future of AI in healthcare is about more than replacing human expertise but augmenting it. Combining AI systems like KARA and ChatGPT with human oversight could offer a robust diagnostic tool, maximizing the strengths of both artificial intelligence and human judgment. As AI continues to evolve, its integration into healthcare systems must be approached thoughtfully, ensuring that it supports and enhances the work of medical professionals for the betterment of patient outcomes.

## Figures and Tables

**Figure 1 diagnostics-14-00090-f001:**
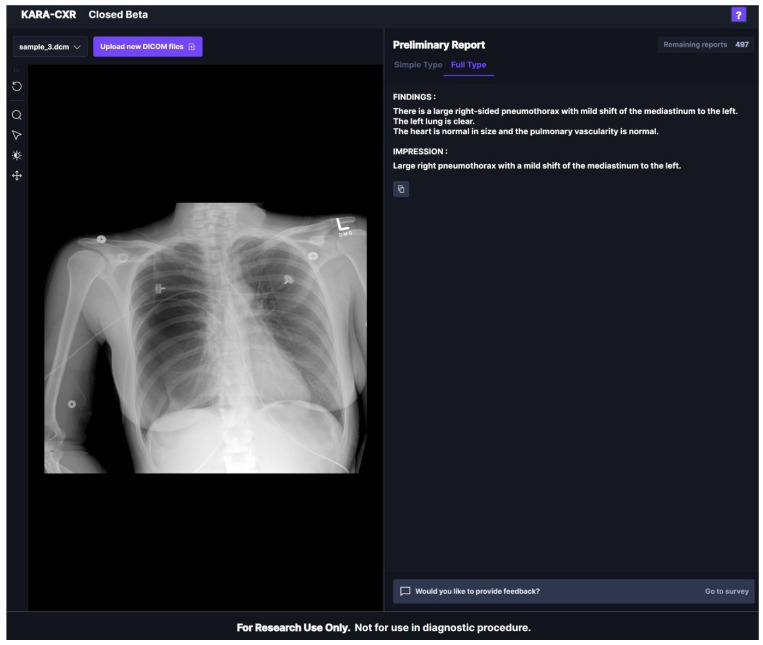
The appearance of KARA-CXR operating in a web browser. When a DICOM file is uploaded, findings and corresponding impressions are displayed on the right-side tab.

**Figure 2 diagnostics-14-00090-f002:**
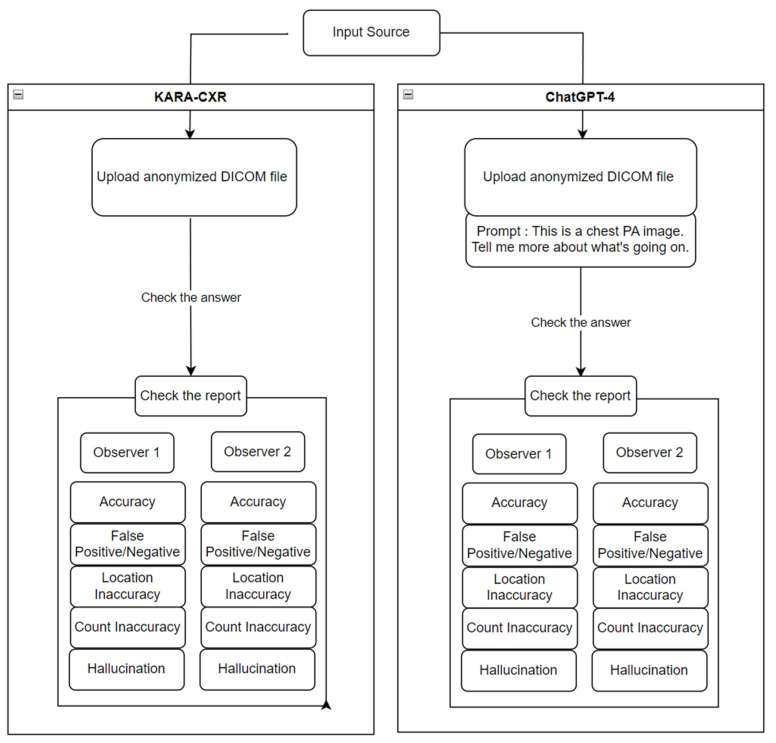
Schematic of data input and analysis process.

**Figure 3 diagnostics-14-00090-f003:**
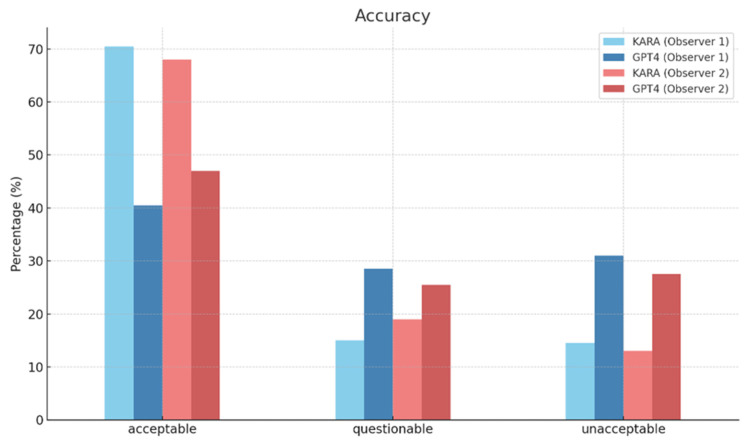
Diagnostic accuracy of KARA-CXR and GPT4.

**Figure 4 diagnostics-14-00090-f004:**
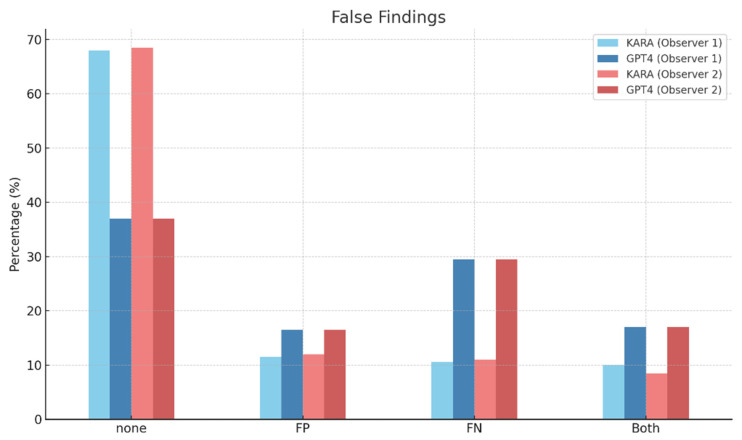
False findings of KARA-CXR and GPT4.

**Figure 5 diagnostics-14-00090-f005:**
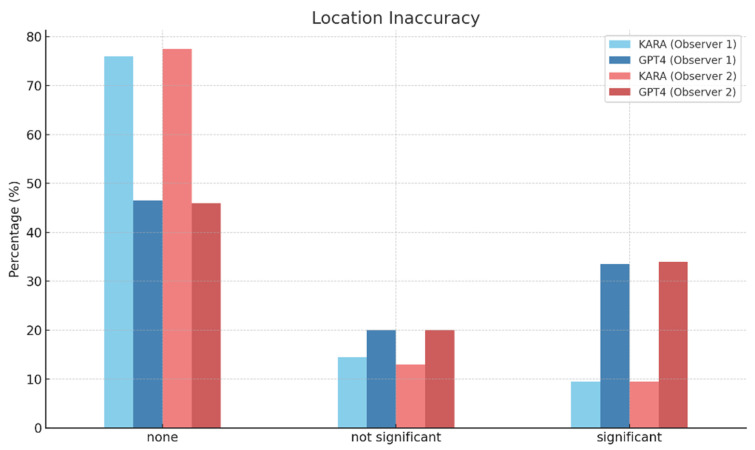
Location inaccuracy of KARA-CXR and GPT4.

**Figure 6 diagnostics-14-00090-f006:**
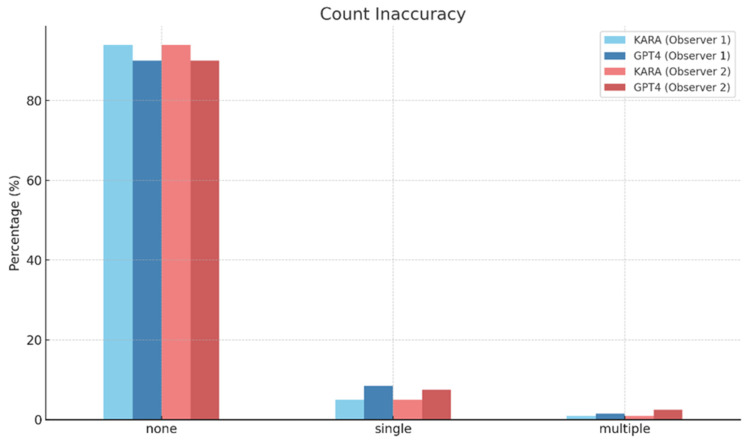
Count inaccuracy of KARA-CXR and GPT4.

**Figure 7 diagnostics-14-00090-f007:**
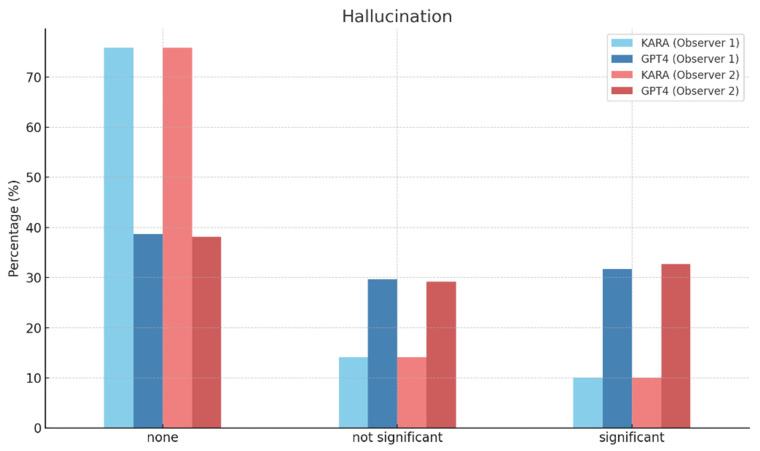
Hallucination of KARA-CXR and GPT4.

**Figure 8 diagnostics-14-00090-f008:**
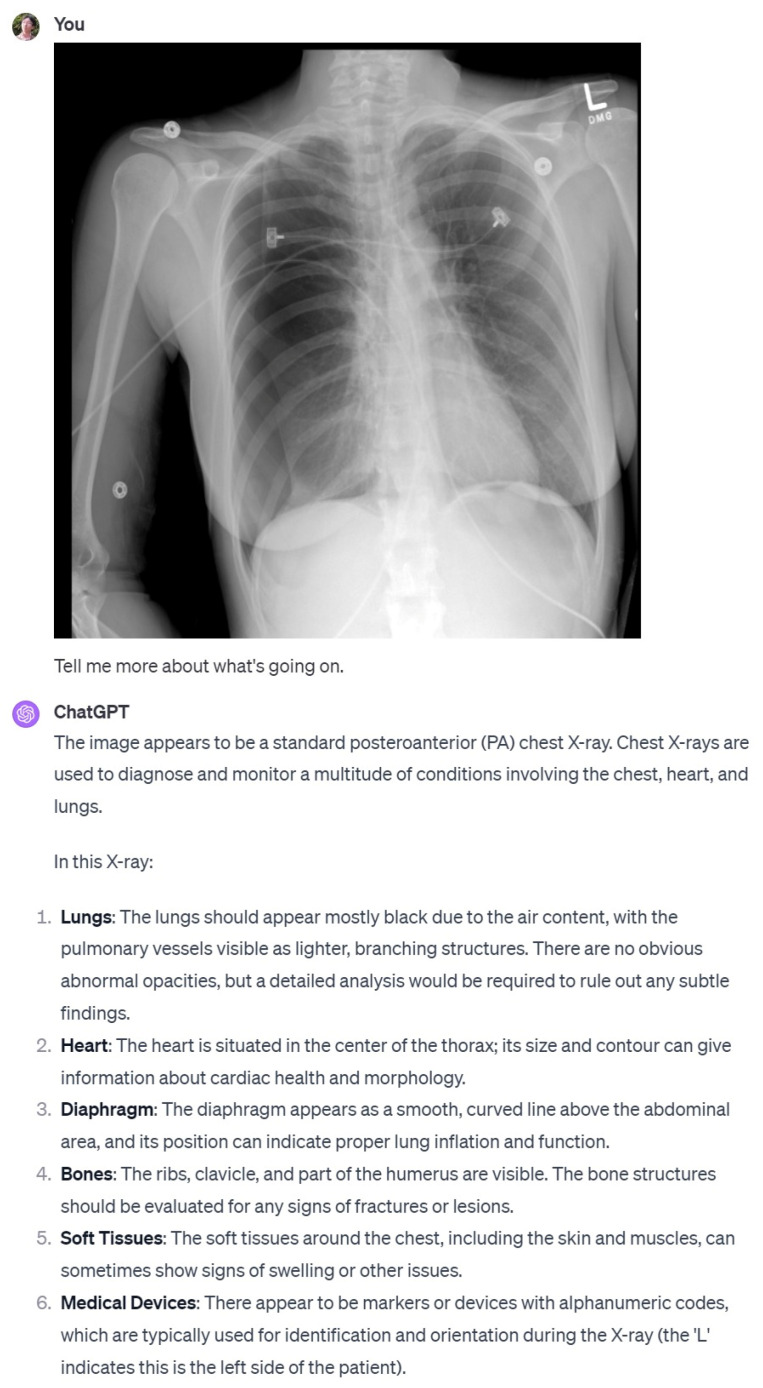
A case where ChatGPT misidentified pneumothorax as normal.

**Table 1 diagnostics-14-00090-t001:** The qualitative factors with which to evaluate the quality of the readings.

Assessment		Description
Accuracy	Acceptable	The reading is accurate and clinically useful.
	Questionable	There are errors in the reading, but it retains some clinical usability.
	Unacceptable	There are significant errors in the reading, rendering it clinically useless.
False Findings	None	There are no false findings.
	False Positive (FP)	The reading includes a false positive.
	False Negative (FN)	The reading includes a false negative.
	Both	The reading has both false positives and false negatives.
Location Inaccuracy	None	There is no location inaccuracy.
	Not significant	The location of lesions is inaccurately identified, but it does not significantly affect clinical judgment.
	Significant	The location of lesions is inaccurately identified, and it severely affects clinical judgment.
Count Inaccuracy	None	There is no count inaccuracy.
	Single	The count of lesions is inaccurate, but single error is noted.
	Multiple	The count of lesions is incorrect and multiple count errors of lesion are seen.
Hallucination	None	There are no hallucinations in the reading.
	Not significant	Hallucinations are present but do not significantly affect clinical judgment.
	Significant	Hallucinations are present and significantly affect clinical judgment.

## Data Availability

The data presented in this study are available on request from the corresponding author. The data are not publicly available due to need approval from the affiliated institution’s DRB (Data review board) is required for disclosure or export.

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
