# Peer review of "Validation of a Deep Learning Chest X-ray Interpretation Model: Integrating Large-Scale AI and Large Language Models for Comparative Analysis with ChatGPT"

_diagnostics, 2023, doi:10.3390/diagnostics14010090_

Round 1

Reviewer 1 Report

Comments and Suggestions for Authors

The paper titled "Advanced Validation of a Deep Learning Chest X-ray Interpretation Model: Integrating Large-Scale AI and Large Language Models for Comparative Analysis with ChatGPT" presents an interesting study that evaluates the diagnostic accuracy of two advanced technologies, KARA-CXR and ChatGPT, in interpreting chest X-ray images. While the study addresses a relevant and promising area of research, there are a few points to consider for further clarity and rigor:

1. The paper should provide a more detailed description of the methodology used for evaluating the diagnostic accuracy of KARA-CXR and ChatGPT. Specifically, how were the qualitative factors (Accuracy, False Findings, Location Inaccuracies, Count Inaccuracies, Hallucination) measured or assessed? A clear and well-defined methodology is crucial for readers to understand the study's approach and results.

2. The paper mentions that 2000 chest X-ray images were randomly selected from a single institution. It would be beneficial to provide more information about the dataset, such as patient demographics, inclusion/exclusion criteria, and any potential biases in the dataset. Additionally, transparency in how the images were labeled (e.g., by radiologists) is essential for understanding the ground truth.

3. Given the importance of AI in medical diagnostics, it's crucial to discuss the ethical and clinical implications of the study's findings. How might the integration of KARA-CXR and ChatGPT impact clinical practice, radiologists, and patient care? Any potential limitations or challenges in implementation should be addressed.

4. The paper mentions the need for transparency in AI algorithms and extensive clinical validation. Expanding on these points with concrete recommendations or insights into addressing these challenges would add depth to the discussion section.

5. Ensure that the paper includes appropriate citations and references to related work in the field of AI in medical imaging. This will help readers understand the context and existing research in this area.

6. The abstract should provide a concise summary of the paper, including key findings, methodology, and implications. Currently, it lacks some details about the methodology and results.

Addressing these points would strengthen the paper and provide a more comprehensive and informative contribution to the field of AI in medical imaging. It's an exciting area of research, and a well-documented study can have a significant impact on improving healthcare outcomes.

Author Response

First, we would like to thank the reviewers for their in-depth reviews. We've responded to your points and made corrections, which we hope to meet the reviewer's intent.

In addition, KARA-CXR, which we used in this paper, is currently in open beta and can be accessed and utilized on the web. (https://karacxr.ai) We hope that you can use it as a reference to understand our research.

# Reviewer 1

  1. The paper should provide a more detailed description of the methodology used for evaluating the diagnostic accuracy of KARA-CXR and ChatGPT. Specifically, how were the qualitative factors (Accuracy, False Findings, Location Inaccuracies, Count Inaccuracies, Hallucination) measured or assessed? A clear and well-defined methodology is crucial for readers to understand the study's approach and results.

A: Thanks for the good points. Due to a submission mistake, a different table was submitted in the initial draft. We have tabulated the definitions of the factors we used to evaluate diagnostic accuracy.

  1. The paper mentions that 2000 chest X-ray images were randomly selected from a single institution. It would be beneficial to provide more information about the dataset, such as patient demographics, inclusion/exclusion criteria, and any potential biases in the dataset. Additionally, transparency in how the images were labeled (e.g., by radiologists) is essential for understanding the ground truth.

A: Thanks for the excellent point. To show the ethnicity component, we indicated the location of the hospital organization (Republic of Korea), the sex ratio of the patients, and the median age. In addition to the previously mentioned exclusion criteria, we have also stated that we excluded images captured on film and with portable equipment. Finally, we indicate that the corresponding author (Ro Woon, Lee) decided which images to include in the dataset.

  1. Given the importance of AI in medical diagnostics, it's crucial to discuss the ethical and clinical implications of the study's findings. How might the integration of KARA-CXR and ChatGPT impact clinical practice, radiologists, and patient care? Any potential limitations or challenges in implementation should be addressed.

A: The study aimed to compare the diagnostic accuracy of chatGPT, and KARA-CXR developed based on language models. Indeed, as mentioned in the discussion, the most problematic aspect of language model-based solutions is the unpredictable "false positives." In this regard, we noted that [Despite the impressive capabilities of AI in healthcare, such as KARA-CXR and ChatGPT, hallucinations can cause serious problems in real-world clinical applications of AI. Such hallucinations may be of minimal consequence in casual conversation or other contexts but can pose significant risks when applied to the healthcare sector, where accuracy and reliability are of paramount importance. Misinformation in the medical domain can lead to severe health consequences on patient care and outcomes. The accuracy and reliability of information provided by language models can be a matter of life or death. They pose real-life risks, as they could potentially affect healthcare decisions, diagnosis, and treatment plans. Hence, the development of methods to evaluate and mitigate such hallucinations is not just of academic interest but of practical importance.]

In fact, among the ethical issues that may arise from such AI solutions, the most important one is that judgments about patients may rely on the AI's judgment without going through the doctor's thoughts. We have added the following sentence to this part.

[In addition to these concerns, it's essential to keep in mind that even if an AI-powered diagnostic solution is highly accurate, the final judgment should still be made by a medical professional - a doctor.]

  1. The paper mentions the need for transparency in AI algorithms and extensive clinical validation. Expanding on these points with concrete recommendations or insights into addressing these challenges would add depth to the discussion section.

A: Good point by the reviewer. One of the problems with these solutions (SaMD, Software as Medical Device) is that we don't have clear regulations yet for clinical validation, licensing, in addition to transparency of algorithms. Based on the reviewer's point, we have added a paragraph as follows.

[Although this paper evaluates the diagnostic accuracy of two potential SaMDs, chatGPT and KARA-CXR, one of the limitations is that there needs to be a clear rationale or recommendation for evaluating or approving such software, legally or within the aca-demic community. The limitation in approving software for medical use (SaMD) stems from the need for a clear definition of SaMD, which makes it difficult to create standards and regulations for its development and implementation [28]. Without clear boundaries, there are risks to patient safety because not all components potentially impacting SaMD are covered by regulations [29]. This lack of clarity also affects innovation and design in the field of SaMD, as new technology applications that support healthcare monitoring and service delivery may need to be more effectively regulated [30]. We believe that gradually, along with software development, we will need to establish factors and reg-ulations that will define the clinical accuracy and safety of these SaMDs.]

  1. Ensure that the paper includes appropriate citations and references to related work in the field of AI in medical imaging. This will help readers understand the context and existing research in this area.

A: That is a good point by the reviewer. There are many papers on chest X-ray diagnosis based on traditional labeling and machine learning.

However, chatGPT, a language model, has recently been able to analyze images (including medical images) through multimodal features, and KARA-CXR is the first attempt based on a language model. The research and development of such language model-based diagnostic solutions is still in its infancy, and we believe that more research should be continued in the future, including this paper.

Please consider that the use of chatGPT and KARA-CXR in medical imaging AI is a new attempt with no similar studies.

  1. The abstract should provide a concise summary of the paper, including key findings, methodology, and implications. Currently, it lacks some details about the methodology and results.

A: We have reorganized the abstract based on the reviewer's good points. Thank you for the review.

Reviewer 2 Report

Comments and Suggestions for Authors

The topic selected for this study is interesting for those who are into the use of AI in healthcare, particularly in radiology practices. However, the work needs significant improvements in terms of study design, scientific writing, citation, etc.

The entire text needs revision, and here are some of the point by point comments.

- As the main theme of the work is about the comparison of the ChatGPT and KARA, I question the use of "Advanced" in the title as it seems unnecessary. Could you please clarify or consider modifying it?

- However, ChatGPT's performance, though slightly inferior, was notably impressive, ...

 Please clarify what is meant by "impressive" as it is a strong term for scientific work. The results indicate errors in obvious cases, suggesting that "impressive" may not be the most accurate descriptor.

- In conclusion, the study underscores the transformative potential ...

 Again here "Transformative" is a strong term. Please ensure it accurately reflects the study's contributions or consider a more precise term.

- For me the 2nd and 3rd paragraphs of the introduction seem identical, with the abbreviation LLMs introduced in both. Consider consolidating them or clearly differentiating their content.

- However, using LLMs also raises ethical and practical challenges...

 The discussion on ethical and practical challenges of LLMs seems disconnected from the main focus of the paper. If this is not central to the study, consider relocating or omitting it to maintain focus.

- However, caution is needed as LLMs can carry certain risks...

Mentioning the risks of LLMs seems out of place here.

- KARA-CXR (Kakaobrain Artificial Radiology Assistance-Chest X-ray) is an innovative medical technology...

The word "revolutionizing" is quite strong. Unless the technology is a definitive game-changer, which I don’t think to be the case. Consider a term that accurately represents its current impact.

-  Please include references in the third paragraph to support the statements made.

-  The fifth paragraph lacks clarity and flow. Please revise it for better coherence and readability.

- Anonymized DICOM files were uploaded to both KARA-CXR and ChatGPT...

 Please provide complete information about the software, including the manufacturer and location. The figure shows it is not a commercial software. If that is the case, so I don’t see any point in highlighting the software with words like revolutionary.

- 'This is a chest PA image. Tell me more about what is going on?'...

 Is this the prompt that is use to task the chatGPT to do diagnosis?! The entire study hinges on the output of this prompt. How do you ensure this prompt is sufficient? For a technical text a more detailed prompt should have been used.

- ChatGPT was used in its paid version, GPT-4V...

 If the images are anonymized before being fed to the model, clarify the necessity of disabling the 'Chat history & training' option. Also, address potential data leakage during processing. I don’t think this adds any layer of protection to the data, as leakage can happen during the processing the images not just after it is done and saed on the cloud.

- 2 comment regarding table 1:

 Firstly, table captions typically appear above the table, not below. Secondly, clarify the criteria for acceptance and how observers reported their accuracy assessments.

- Two readers (R.W.L, K.H.L.)...

 Please clarify the meaning of the initials. If they represent observer names, consider removing them for a more scientific approach.

- The order of evaluation of the images was performed randomly...

  Please revise this sentence for better grammar and clarity.

- The analysis of the data was performed in Python (version 3.12).

 Please make it clear what "processing" entails. If it refers to only anonymization or another specific process, state it explicitly.

- Comments on figure 3:

  Please define the terms "acceptable," "questionable," and "unacceptable" within the text.

- The rise of medical image reading with large language models has gained significant attention...

 Provide a reference for this claim to strengthen the statement.

- The ChatGPT model has explicitly been highlighted...

Include references to substantiate this statement and clarify what is specifically meant.

- The goal is to bridge the gap...

 This statement is vague. Include references or specify whether this pertains to this study specifically or the general research area.

- ChatGPT, powered by the GPT-4V architecture...

Provide references to support this claim about the potential of ChatGPT in the medical field. I don’t think this can be true as it has not a certified product for medical applications.

- It should be integrated into clinical workflows...

How this is possible without regulatory checks and other stuff?

- This study underscores the potential of AI in revolutionizing medical diagnostics...

 Reconsider this statement, as "revolutionizing" implies a level of impact that may not be supported by the study's findings.

- Lijiao, Zhang., Kai, Sun., Akshay, T., Jagadeesh., Deepa, Gupta., Vibhor, Gupta., Yike, Guo. (2023)...

Please specify the journal where this work is published for reference.

Comments on the Quality of English Language

generally OK but some revision is needed.

Author Response

First, we would like to thank the reviewers for their in-depth reviews. We've responded to your points and made corrections, which we hope to meet the reviewer's intent.

In addition, KARA-CXR, which we used in this paper, is currently in open beta and can be accessed and utilized on the web. (https://karacxr.ai) We hope that you can use it as a reference to understand our research.

# Reviewer 2

First, we would like to thank Reviewer 2 for your detailed and insightful review of this paper, which is subject to some revisions.

  1. As the main theme of the work is about the comparison of the ChatGPT and KARA, I question the use of "Advanced" in the title as it seems unnecessary. Could you please clarify or consider modifying it?

A: We've removed the term "advanced" based on a good point made by a reviewer.

  1. However, ChatGPT's performance, though slightly inferior, was notably impressive, ...

 Please clarify what is meant by "impressive" as it is a strong term for scientific work. The results indicate errors in obvious cases, suggesting that "impressive" may not be the most accurate descriptor.

A: Thanks for your good point. We have rewritten the abstract to take into account the other reviewers and your points in paragraphs 2 and 3.

  1. In conclusion, the study underscores the transformative potential ...

 Again here "Transformative" is a strong term. Please ensure it accurately reflects the study's contributions or consider a more precise term.

A: Thanks for your good point. We have rewritten the abstract to take into account the other reviewers and your points in paragraphs 2 and 3.

  1. For me the 2nd and 3rd paragraphs of the introduction seem identical, with the abbreviation LLMs introduced in both. Consider consolidating them or clearly differentiating their content.

A: Thank you for your good points. I've cleaned up the acronyms for clearly differentiating our content.

  1. However, using LLMs also raises ethical and practical challenges...

 The discussion on ethical and practical challenges of LLMs seems disconnected from the main focus of the paper. If this is not central to the study, consider relocating or omitting it to maintain focus.

A: As pointed out by the reviewer, this part has been omitted.

  1. However, caution is needed as LLMs can carry certain risks...

Mentioning the risks of LLMs seems out of place here.

A: As pointed out by the reviewer, this part has been omitted.

  1. KARA-CXR (Kakaobrain Artificial Radiology Assistance-Chest X-ray) is an innovative medical technology...

The word "revolutionizing" is quite strong. Unless the technology is a definitive game-changer, which I don’t think to be the case. Consider a term that accurately represents its current impact.

A: Based on the reviewer's advice, we've revised the sentence as follows

[KakaoBrain Artificial Radiology Assisted Chest X-ray (KARA-CXR) is a new medical technology that helps in radiological diagnosis.]

  1. Please include references in the third paragraph to support the statements made.

A: As pointed out by the reviewer, we have checked the references in the sentences and added a reference to the last sentence as follows

[LLMs, like ChatGPT, can accelerate the creation of clinical practice guidelines by quickly searching and selecting evidence from numerous databases [6].]

  1. The fifth paragraph lacks clarity and flow. Please revise it for better coherence and readability.

A: In response to the reviewer's comments, we've reorganized the somewhat exaggerated description of chatGPT's utility and tried to make it clearer and more flowing.

[Based on the GPT-4V architecture, ChatGPT has potential in the medical field, especially for interpreting chest X-ray images. This language model can analyze medical images, including chest X-ray data, to generate human-like reading reports. Although not yet available for clinical use, by providing a general interpretation of chest X-rays, ChatGPT has the potential to improve the diagnostic process, especially in settings with limited access to radiology expertise [8].]

  1. Anonymized DICOM files were uploaded to both KARA-CXR and ChatGPT...

 Please provide complete information about the software, including the manufacturer and location. The figure shows it is not a commercial software. If that is the case, so I don’t see any point in highlighting the software with words like revolutionary.

A: Thanks for the reviewer's good point. We have rewritten the sentence to include the manufacturer and location of the software in question.

[Anonymized DICOM files were uploaded to both KARA-CXR (Kakaobrain, Seoul, Republic of Korea) and ChatGPT (OpenAI, San Francisco, CA).]

  1. 'This is a chest PA image. Tell me more about what is going on?'...

 Is this the prompt that is use to task the chatGPT to do diagnosis?! The entire study hinges on the output of this prompt. How do you ensure this prompt is sufficient? For a technical text a more detailed prompt should have been used.

A: That's a very good point. In general, chatGPT seems to be able to answer all questions, but OpenAI's policy is that it cannot answer prompts about specialized knowledge (medical, legal, criminal, etc.). This was somewhat circumvented in earlier versions, but has become increasingly restricted with each upgrade.

However, as utilized in this thesis, bypassing the prompts can provide information about the desired specialized knowledge, which is sometimes referred to as "jailbreaking". A limitation of our study is that we were unable to ask for direct reading prompts. We attach an example of refusal when requesting a reading as a reference figure.

  1. ChatGPT was used in its paid version, GPT-4V...

 If the images are anonymized before being fed to the model, clarify the necessity of disabling the 'Chat history & training' option. Also, address potential data leakage during processing. I don’t think this adds any layer of protection to the data, as leakage can happen during the processing the images not just after it is done and saed on the cloud.

A: In response to the reviewer's comments, we've added the following language to describe the extent to which data is anonymized and the options available to protect privacy.

[To anonymize medical images for analysis in ChatGPT and KARA-CXR, we removed all identifiable patient information to comply with the privacy and confidentiality standards set forth by the Health Insurance Portability and Accountability Act (HIPAA). Anonymization involved removing details such as patient names, dates of birth, medical record numbers, and other unique identifiers from the images.

Even after anonymization, we further enhanced privacy by turning off the "Chat History and Training" option in ChatGPT. This setting ensures that conversations and images shared during a session are not used for further training of the AI model or accessed in future sessions. This is a precautionary measure to ensure that residual or in-direct information is not used in ways that could compromise patient confidentiality.

In KARA-CXR, KARA-CXR, a cloud-based analysis system, immediately deleted the input DICOM data after analysis for personal information protection.]

  1. 2 comment regarding table 1:

 Firstly, table captions typically appear above the table, not below. Secondly, clarify the criteria for acceptance and how observers reported their accuracy assessments.

A: In response to a reviewer's comment, we've added a definition of the criteria for evaluating accuracy. We've also corrected the positioning of the caption.

  1. Two readers (R.W.L, K.H.L.)...

 Please clarify the meaning of the initials. If they represent observer names, consider removing them for a more scientific approach.

A: We have included the full names of the two observers (radiologist, author) as noted by the reviewer.

  1. The order of evaluation of the images was performed randomly...

  Please revise this sentence for better grammar and clarity.

A: As pointed out by the reviewer, we have modified the sentence as follows

[We evaluated the interpretation results of each model for chest X-ray images and recorded the evaluation results according to the case numbers of the anonymized images.]

  1. The analysis of the data was performed in Python (version 3.12).

 Please make it clear what "processing" entails. If it refers to only anonymization or another specific process, state it explicitly.

A: Due to the simplicity of the statistical analysis in this study (calculating percentages, inter-observer agreement), we used Python for the statistical analysis. We have modified the sentence to clarify this.

[The statistical analysis of the data was performed in Python (version 3.12).]

  1. Comments on figure 3:

  Please define the terms "acceptable," "questionable," and "unacceptable" within the text.

A: Thanks for the good points. Due to a submission mistake, a different table (table 1) was submitted in the initial draft. We have tabulated the definitions of the factors we used to evaluate diagnostic accuracy on new table 1.

We believe that the reader will be able to understand the concept of these definitions better than before this part, and we deeply apologize for not providing clear definitions in the first draft.

  1. The rise of medical image reading with large language models has gained significant attention...

 Provide a reference for this claim to strengthen the statement.

A: Thanks for the reviewer's good point. Following the reviewer's advice, we have added 1 paper as references.

[[18] Samriddhi, Srivastav., Rashi, Chandrakar., Sristy, Agrawal., Arpita, Jaiswal., Roshan, Prasad., Mayur, Wanjari. (2023). ChatGPT in Radiology: The Advantages and Limitations of Artificial Intelligence for Medical Imaging Diagnosis. Cureus, doi: 10.7759/cureus.41435]

  1. The ChatGPT model has explicitly been highlighted...

Include references to substantiate this statement and clarify what is specifically meant.

A: The wording is somewhat ambiguous, and we believe it is redundant with the first sentence of the paragraph (The rise of medical image..). Consequently, we have removed this sentence.

  1. The goal is to bridge the gap...

 This statement is vague. Include references or specify whether this pertains to this study specifically or the general research area.

A: This is more of a personal opinion of the author. I've removed it to eliminate any ambiguity. Thanks for the good point.

  1. ChatGPT, powered by the GPT-4V architecture...

Provide references to support this claim about the potential of ChatGPT in the medical field. I don’t think this can be true as it has not a certified product for medical applications.

A: The reviewer's advice is correct. Unlike KARA-CXR, which is being developed for medical approval, chatGPT is a general-purpose macrolanguage model and cannot guarantee medical expertise. Based on the reviewers' advice and the authors' judgment, we have revised the paragraph overall.

[In this study, ChatGPT based on GPT-4V architecture showed some potential in interpreting chest X-ray images but also revealed some limitations. ChatGPT can generate human-like diagnostic reports based on chest X-ray data through extensive reinforcement learning on the medical text and imaging data included during development. However, due to the limitations of reinforcement learning based on information openly available on the internet, we must recognize that the data generated by ChatGPT does not guarantee medical expertise. In conclusion, it is essential to note that ChatGPT is not a substitute for professional medical advice, diagnosis, or treatment.]

  1. It should be integrated into clinical workflows...

How this is possible without regulatory checks and other stuff?

A: The reviewer's advice is correct. It is true that there is no LLM SaMD that has yet been approved for licensure by a regulatory authority. We have removed this sentence as it is a bit of a leap.

  1. This study underscores the potential of AI in revolutionizing medical diagnostics...

 Reconsider this statement, as "revolutionizing" implies a level of impact that may not be supported by the study's findings.

A: As pointed out by the reviewer, we have modified the sentence as follows

[This study underscores the potential of AI in improving medical diagnostic processes, with specific emphasis on chest X-ray interpretation.]

  1. Lijiao, Zhang., Kai, Sun., Akshay, T., Jagadeesh., Deepa, Gupta., Vibhor, Gupta., Yike, Guo. (2023)...

Please specify the journal where this work is published for reference.

A: This article is from the ArXiv (https://arxiv.org/abs/2307.08152) and we have modified the citations.

[Lijiao, Zhang., Kai, Sun., Akshay, T., Jagadeesh., Deepa, Gupta., Vibhor, Gupta., Yike, Guo. (2023). The Potential and Pitfalls of using a Large Language Model such as ChatGPT or GPT-4 as a Clinical Assistant. arXiv.org, doi: 10.48550/arXiv.2307.08152]

Round 2

Reviewer 1 Report

Comments and Suggestions for Authors

The paper can be accepted in the current form

Author Response

We greatly appreciate the hard work of our reviewers.

Reviewer 2 Report

Comments and Suggestions for Authors

The revised manuscript presents an enhanced scientific tone and overall improvement from the initial version. However, there are critical points that needs to be addressed. Firstly, the paper must explicitly address a limitation within the discussion section: the inability of ChatGPT to interpret X-ray images using specific prompts. As noted in authors reply to the comments, ChatGPT ceases to function when tasked specifically with analyzing X-ray images, providing a warning against assuming the role of a radiologist. This is a fundamental limitation and should be clearly expressed in the manuscript. Also, in the methodology section, this rationale behind the selection of a general prompt needs to be explained in detail. This detailing illustrates that with certain prompts, ChatGPT refrains from generating results. This detail is vital to ensure the replicability and comprehension of the study's methodology. 

Regarding the acronym of the radiologists' names, it appears there was a misunderstanding of the initial comment. Typically, the inclusion of a radiologist's name does not enhance the value of the manuscript and should remain confidential. The suggestion for the authors was to omit rather than fully include the radiologist's name, as the use of acronyms in the previous manuscript version was ambiguous and potentially misleading. 

Author Response

# Point to point response to reviewer 2

I would like to express my deep gratitude for the thorough review provided by the reviewer during the evaluation process.

  • Firstly, the paper must explicitly address a limitation within the discussion section: the inability of ChatGPT to interpret X-ray images using specific prompts. As noted in authors reply to the comments, ChatGPT ceases to function when tasked specifically with analyzing X-ray images, providing a warning against assuming the role of a radiologist. This is a fundamental limitation and should be clearly expressed in the manuscript.

A: Thank you for the reviewer's good point. After we did the initial study design and selected the images, we were surprised to find that when we asked ChatGPT to read the chest X-ray, the reading was unavailable. To address this, we used a detouring non-directive prompt to achieve our study objectives. We have added the following paragraph to the Discussion section to address this limitation and added a reference to ChatGPT's usage policy.

[In our exploration of ChatGPT's application to medical imaging, particularly in chest X-ray interpretation, a notable limitation emerged, meriting explicit mention. ChatGPT, in its current design, is programmed to refuse direct requests for professional interpretation of medical images, such as X-rays [8]. This usage policy and ethical boundary, built into ChatGPT to avoid the non-professional practice of medicine, significantly impacts its clinical application in this context. In the initial process of our study, we observed that direct prompts requesting chest X-ray interpretation were consistently declined by ChatGPT, aligning with its programming to avoid assuming the role of a radiologist or other medical professional. This limitation is critical to understand for any future research utilizing ChatGPT or similar language models in medical image interpretation.]

[8] OpenAI. (2023). Usage policies. Retrieved December 28, 2023, from https://openai.com/policies/usage-policies.

  • Also, in the methodology section, this rationale behind the selection of a general prompt needs to be explained in detail. This detailing illustrates that with certain prompts, ChatGPT refrains from generating results. This detail is vital to ensure the replicability and comprehension of the study's methodology. 

A: In response to the reviewer's point, we've detailed our rationale for choosing the prompts as follows and also added a reference to ChatGPT's usage policy.

[In the case of ChatGPT, to get the right results for our research, we first chose the prompt to be entered into ChatGPT. ChatGPT is designed with guidelines that prevent it from providing professional interpretations or diagnoses, especially in contexts requiring specialized expertise, such as medical imaging, including chest X-rays [8]. To over-come the limitations of this large language model, we employed a carefully crafted, non-directive bypass prompt: 'This is a chest PA image. Tell me more about what is going on?' This prompt was strategically chosen to navigate ChatGPT's usage policies and ethical constraints, allowing us to obtain a chest x-ray reading from ChatGPT.]

[8] OpenAI. (2023). Usage policies. Retrieved December 28, 2023, from https://openai.com/policies/usage-policies.

  1. Regarding the acronym of the radiologists' names, it appears there was a misunderstanding of the initial comment. Typically, the inclusion of a radiologist's name does not enhance the value of the manuscript and should remain confidential. The suggestion for the authors was to omit rather than fully include the radiologist's name, as the use of acronyms in the previous manuscript version was ambiguous and potentially misleading. 

A: Thank you for your insightful comments. Following your recommendation, the name of the radiologist has been removed.
